# Aspirin Therapy, Cognitive Impairment, and Dementia—A Review

Elizabeth H. Thong, Edward C. Y. Lee, Choi-Ying Yun, Tony Y. W. Li and Ching-Hui Sia *

Department of Cardiology, National University Heart Centre, 1E Kent Ridge Road, National University Health System Tower Block, Level 9, Singapore 119228, Singapore
* Correspondence: ching_hui_sia@nuhs.edu.sg; Tel.: +65-67795555

**Abstract:** Background: Dementia is associated with a greater burden of cardiovascular risk factors. There is a significant vascular contribution to dementia, and aspirin may play a role in targeting this vascular dysregulation via its anti-inflammatory and antiplatelet effects. We provide an overview of the effects of aspirin therapy on the prevention of dementia and cognitive decline in patients with or without dementia and/or cognitive impairment. Methods: We performed a search for studies enrolling adults with or without dementia or MCI and comparing aspirin with placebo, usual care, or active control with respect to cognitive outcomes. Results: We describe aspirin's effects on the primary prevention of cognitive impairment and various subtypes of dementia, as well as its role in cognitive decline in certain subsets of patients, including those with cerebral small vessel disease (CVSD), coronary heart disease (CHD), and gender differences. Overall, the benefits of aspirin in preventing dementia and cognitive decline remain inconclusive. The majority of cohort studies investigating aspirin's role in preventing cognitive decline or dementia looked promising, but this was not supported in most randomised controlled trials. However, aspirin may still be beneficial in certain subgroups of patients (such as CHD, VD, and CSVD) and warrants further investigation.

**Keywords:** aspirin; acetylsalicylic acid; ASA; cognitive impairment; cognitive decline; dementia; Alzheimer's dementia; vascular dementia; cerebral small vessel disease; gender; sex; cerebral haemorrhage

## 1. Introduction

Dementia is a syndrome caused by chronic or progressive disease of the brain, associated with impaired higher cortical functions, and interfering with daily functioning and quality of life [1]. Mild cognitive impairment (MCI) is the intermediate state between normal cognition and dementia, with impaired higher cortical functions but preserved functional abilities [2]. The pathology of dementia depends on its aetiology. In AD, key features include cerebral atrophy, particularly in the medial temporal lobe and hippocampus. There are also extracellular β-amyloid plaques and neurofibrillary tangles deposited throughout the cerebral cortex, which accumulate in increasing density as the dementia progresses [3]. The presence of cerebral amyloid angiopathy itself is also a risk factor for stroke in the elderly, with increased risk of ischaemic and haemorrhagic stroke [3]. In dementia caused by cerebrovascular disease, microinfarcts in the cerebral cortex and degeneration of small blood vessels are associated with enlarged periventricular spaces, as well as old haemorrhages, cerebral amyloid angiopathy, and hippocampal sclerosis [3]. The pathophysiology of mild cognitive impairment (MCI) is not well established, as it does not have a clear pathological or clinical course. Some patients present with vascular symptoms and progress to AD; indeed, β-amyloid plaques and senile plaques are a histological hallmark of MCI. The pathology of MCI also includes a wide range of cellular dysfunction, initiation of neuroplastic response, and neuronal disconnection in the central nervous system, in addition to the pathology of senile plaques and neurofibrillary tangles [4]. The

extent of each factor's contribution also depends on the clinical subtype of MCI, and cannot be simply attributed to a single pathological process.

On the whole, dementia is associated with a greater burden of cardiovascular risk factors, including diabetes mellitus, smoking, hypertension, and metabolic syndrome [5]. As such, there is a significant vascular contribution to dementia, with the presence of vascular dysregulation doubling the chance that the neurodegenerative pathology will evolve into dementia [6]. Indeed, all major dementias involve some degree of vascular dysregulation, ranging from 61% in frontotemporal dementia to 80% in Alzheimer's disease (AD) [6]. As such, aspirin may play a role in targeting this vascular dysregulation. Aspirin is one of the most commonly used drugs for primary and secondary prevention of cardiovascular disease [7]. It is a non-competitive, irreversible inhibitor of cyclooxygenase (COX)-1, -2, and -3. At low doses of 75–300 mg/day, its primary effects are in COX-1 inhibition and subsequent thromboxane A2 synthesis [8]. This reduces platelet aggregation and vasoconstriction, resulting in reduced intravascular clot formation. Neurodegenerative diseases have also been found to have increased expression of COX-1 [9,10], and studies have shown that it is a central part of the neuroinflammatory response [11]. As such, studies have shown that COX-1 inhibition reduces neuroinflammation mediated by β-amyloid proteins and also reduces pro-inflammatory cytokine release (Figure 1) [12,13]. Animal studies on COX-1 inhibition also show its potential role in preventing working memory deficits [14]. The selective COX-1 inhibitor triflusal was shown in a randomised controlled trial to reduce the progression of MCI to dementia [15].

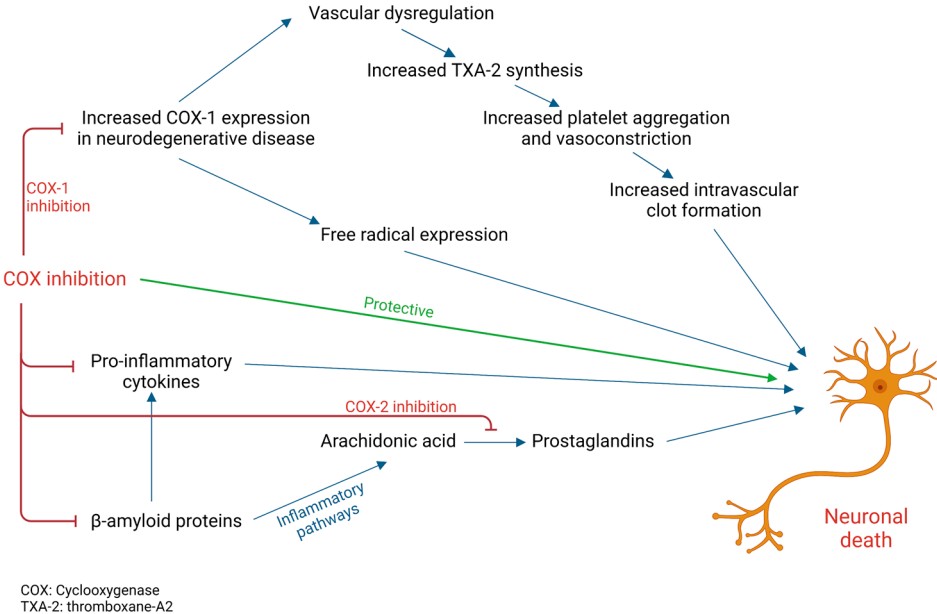

COX: Cyclooxygenase
TXA-2: thromboxane-A2

**Figure 1.** Potential mechanisms of COX inhibition on preventing neural death.

Hence, the COX-1 and -2-inhibitory properties of aspirin and its role in reducing cognitive decline are of interest, with some studies showing that aspirin reduces serum biochemical factors associated with cognitive impairment [16]. However, aspirin could potentially increase the risk of cerebral microbleeds via inhibition of platelet aggregation, which could also increase the risk of cognitive impairment [17,18]. Hence, multiple studies have investigated whether the antiplatelet and anti-inflammatory effects of aspirin can be used in the prevention of dementia and cognitive decline.

Our objective is to provide an overview of the effects of aspirin therapy on the prevention of dementia and cognitive decline in patients with or without dementia and/or cognitive impairment.

## 2. Materials and Methods

We performed a search of MEDLINE, Embase, and Scopus on 2 November 2022 for studies from inception until 2 November 2022, including the following terms: (cognition OR confusion OR cognitive deficit OR cognitive decline OR cognitive impairment OR dementia OR Alzheimer* OR vascular dementia OR neuropsych* test OR neuropsych* deficit OR memory OR neuroimaging) AND (aspirin OR antiplatelet OR acetylsalicylic acid OR cyclooxygenase inhibitor OR acemetacin OR anti-inflammatory). We included studies enrolling adults with or without dementia or MCI and comparing aspirin with placebo, usual care, or active control with respect to cognitive outcomes. The primary outcomes were diagnoses of cognitive impairment or dementia. The included literature consisted of relevant and original research based on titles and abstracts as well as selected full texts. Additional articles were identified from a manual search of the references of relevant reviews. A full-text review was performed for all relevant articles.

## 3. Results

Main trial results are summarised in Table 1.

**Table 1.** Summary table of major trials discussed. AD: Alzheimer's dementia; MMSE: Mini mental state exam; VD: Vascular dementia; CSVD: Cerebral small vessel disease; CHD: Coronary heart disease.

| Effect of Aspirin on Cognitive Impairment and Dementia | | | | |
|---|---|---|---|---|
| **Effect of Aspirin** | **Trial** | **Trial Design** | **Patient Number** | **Outcome** |
| Primary Prevention of Cognitive Impairment and Dementia | ASCEND [19] | Randomised controlled trial | 15,480 | Rates of dementia, cognitive impairment, or confusion were similar to placebo |
| | JPAD [20] | Randomised controlled trial | 2121 | No significant difference between aspirin vs placebo in the prevention of dementia |
| | AAA [21] | Randomised controlled trial | 3350 | Did not show any benefit in preserving cognitive function |
| Aspirin and Cognitive Decline in Alzheimer's Dementia (AD) | ASPREE [22] | Randomised controlled trial | 19,114 | Did not find any significant difference between the aspirin and placebo groups with regards to dementia |
| | AD2000 [23] | Randomised open-label trial | 310 | No association between aspirin and risk of AD |
| | Health in Men [24] | Population-based retrospective cohort study | 3679 | No association between aspirin and risk of AD |
| | Chang CW et al. [25] | Population-based retrospective cohort study | 28,321 | Yes–decreased risk of incident AD |
| | Ferrari C et al. [26] | Cohort study | 160 | Yes–odds of a patient on aspirin having a rapid decline in MMSE over 2 years were lower |
| | Sydney Older Persons Study [27] | Longitudinal study | 647 | Yes–inverse association between aspirin and presence of AD |
| | Alzheimer's Disease Neuroimaging Initiative [28] | Longitudinal study | 1866 | Yes–associated with slower cognitive decline in AD, but not but in patients with normal cognition or mild cognitive impairment |
| Aspirin and Cognitive Decline in Vascular Dementia (VD) | Meyer et al. [29] | Randomised controlled trial | 70 | Yes–associated with better cognitive scores and cerebral perfusion values |
| | ESTHER [30] | Prospective cohort study | 5258 | Yes–aspirin reduced hazard for VD |
| | UK Biobank [31] | Prospective cohort study | 305,394 | Yes–aspirin reduced hazard for VD |
| | Sydney Older Persons Study [27] | Longitudinal study | 647 | No–did not protect against VD |
| | Cardiovascular Health Cognition Study [32] | Prospective cohort study | 3229 | No–did not protect against VD |
| | Canadian Study of Health and Aging [33] | Cohort study | 8623 | Increased risk of VD |
| | Rotterdam study [34] | Cohort study | 6989 | Increased risk of VD |
| Aspirin and Cognitive Decline in Cerebral Small Vessel Disease (CSVD) | CHALLENGE trial [35] | Multicentre, double-blind, randomised controlled trial | 256 | No difference in the effect of cilostazol or aspirin compared to the expected progression of white matter change in CSVD |
| | Silence study [36] | Longitudinal, randomised, double blind controlled trial | 83 | No difference in cognitive decline |

**Table 1.** *Cont.*

| Effect of Aspirin | Trial | Trial Design | Patient Number | Outcome |
|---|---|---|---|---|
| **Effect of Aspirin on Cognitive Impairment and Dementia** | | | | |
| Aspirin and Dementia in Patients with Coronary Heart Disease (CHD) | Meta-analysis [37] of ESTHER [30] and UK Biobank [31] | | | Subjects with pre-existing CHD benefited strongly from low-dose aspirin |
| Aspirin and Dementia in Patients with Late Onset Depression (LOD) | Ya-Hsu Yang et al. [38] | Population-based study | 46,439 | Lower incidence of incident dementia in patients on aspirin |
| Aspirin in Women | JPAD2 [39] | Longitudinal follow up of original JPAD cohort | 2359 | Significant reduction in the risk of dementia was seen in in women but not in men |
| | Kern S et al. [40] | Population-based cohort study | 681 | Significantly less dementia in high cardiovascular risk women who were randomised to the aspirin group |
| | Alzheimer's Disease Neuroimaging Initiative [28] | Longitudinal study | 1866 | Aspirin was associated with slower decline in male AD patients but not female AD patients |
| | Meta-analysis [37] of ESTHER [30] and UK Biobank [31] | | | Aspirin was associated with decreased risk of all cause dementia and VD in males, but not in females |
| | Women's Health Study [41] | Randomised, double blind, placebo-controlled cohort study | 6377 | Women assigned aspirin did not do better than those placebos; some evidence for reduced decline in category (semantic) fluency in the aspirin group; aspirin was associated with less cognitive decline in current smokers and hyperlipidaemia |
| | Kang et al. [42] | Observational study | 13,255 | No association with cognitive decline |
| High Dose Aspirin | Nilsson SE et al. [43] | Longitudinal study | 702 | High-dose aspirin resulted in lower prevalence of AD and better cognitive function |
| | Sydney Older Persons Study [27] | Longitudinal study | 647 | Whilst aspirin reduced risk of development of Alzheimer's disease, there was no difference in outcome between low and high doses |

### 3.1. Primary Prevention of Cognitive Impairment and Dementia

While multiple studies have investigated the effects of aspirin in the primary prevention of cardiovascular disease, only a limited number of trials have reported on its effects on dementia. Previous longitudinal data from the Canadian Health Study [44], along with another randomised controlled trial (RCT) of 1007 elderly patients [45], reported a protective effect of aspirin against cognitive decline. However, other longitudinal studies did not find any differences in outcomes [46], and data from the Cardiovascular Health Study of 3229 patients aged >65 years also reported no protective effect from aspirin [32].

There are a few key RCTs that have investigated the role of aspirin in cognitive decline and dementia. The ASPREE (Aspirin in Reducing Events in the Elderly) RCT enrolled 19,114 dementia-free patients over 70 years old into either aspirin or placebo groups. It was terminated early after 4.7 years of follow-up as there was no benefit with aspirin. It did not find any significant difference between the aspirin and placebo groups with regards to dementia, with aspirin also conferring a higher rate of major haemorrhage [47]. In addition, aspirin did not significantly improve performance in any specific cognitive domains (i.e., memory, psychomotor speed, language, and executive function). Subgroup analysis also did not find that aspirin slowed cognitive decline across subgroups of age, sex, ethnic or racial group, health factors, or prior NSAID use [22].

Similarly, the ASCEND (A Study of Cardiovascular Events iN Diabetes) RCT followed 15,480 patients with diabetes for a mean of 7.4 years after randomising them to aspirin or placebo groups. Like ASPREE, the ASCEND trial found that rates of dementia, cognitive impairment, or confusion were similar across the groups [19]. The third RCT, JPAD (Japanese Primary Prevention of Atherosclerosis with Aspirin for Diabetes trial), involved 2121 patients with type 2 diabetes mellitus, with no significant differences between the aspirin and placebo groups in the prevention of dementia [20,35]. A meta-analysis of these three RCTs yielded a combined hazard ratio of 0.92 (95% CI 0.84–1.01) ($p = 0.09$) [19]. The AAA (Aspirin for Asymptomatic Atherosclerosis Trial) RCT of low-dose aspirin in 3350 men and women aged over 50 years and at moderately increased risk of cardiovascular

disease also failed to show any benefit in preserving cognitive function [21]. A Scottish trial of 3350 patients with moderately increased cardiovascular risk did not find any difference in cognitive function with aspirin [21].

A 2017 meta-analysis of the effects of aspirin on cognitive function also found no evidence for any effect [48]. A more recent meta-analysis in 2021 found that, while cohort studies found that incident dementia was reduced by low-dose aspirin, this was not supported by RCTs [20]. As such, the role of aspirin in preventing overall cognitive decline and dementia is still debatable. Further robust studies are required to ascertain its effectiveness, and examination of the effects of aspirin in specific situations and dementia subtypes is still warranted.

### 3.2. Aspirin and Cognitive Decline in Alzheimer's Disease (AD)

Underlying inflammation has been found to be one of the key pathological processes behind Alzheimer's disease, and cerebrovascular disease burden plays a large role in the onset of vascular dementia [6,49,50]. In Alzheimer's disease (AD), there is characteristic aggregation of neurotoxic peptides in the intra- and extracellular regions of the brain associated with surrounding neuroinflammation [51,52]. The extracellular deposition of amyloid β-peptides, intracellular neurofibrillary tangle deposition, and neuronal loss induce inflammatory pathways that result in accumulation of prostaglandins and other inflammatory mediators [53]. As such, increased inflammatory mediators in the CSF and the brain parenchyma are associated with cognitive impairment [51,52,54,55]. This suggests that persistent neuroinflammation may be a driver of disease progression in neurodegenerative diseases and in AD [51,52]. In addition, neuroinflammation is present throughout the course of AD, from the initial asymptomatic disease phase to the final advanced stages [56]. This suggests that there may be a role of targeted anti-inflammatory therapy in preventing dementia and cognitive decline.

Iturria-Medina et al. conducted a multifactorial data-driven analysis of over 7700 brain images and multiple biomarkers from the Alzheimer's Disease Neuroimaging Initiative. They found that intra-brain vascular dysregulation was involved in the early pathology of late-onset Alzheimer's disease, and early memory deficit was associated with highly abnormal proteins in the vascular system [57]. As such, reducing the risk of vascular dysregulation has the potential to reduce the incidence of both cardiovascular disease and dementia.

There are mixed results on the role of aspirin in reducing AD. The ASPREE double-blind, placebo-controlled RCT of 19,114 participants did not find that aspirin reduced the risk of AD [22]. Similarly, the AD2000 (Aspirin in Alzheimer's disease, a randomised open-label trial) trial of 310 patients with AD and the Health in Men study of 3679 men with cardiovascular disease both found no association between aspirin and the risk of AD [23,24].

In contrast, a population-based retrospective cohort study from Taiwan comprising 28,321 patients with type 2 diabetes mellitus found that low-dose aspirin (40 mg) was associated with a decreased risk of incident AD [25]. Similarly, a cohort study from Italy comprising 160 patients with AD found that the odds of a patient on aspirin having a rapid decline in MMSE scores over 2 years were lower compared to controls (OR 0.34, 95% CI 0.11–0.88) [26]. The Sydney Older Persons Study longitudinal cohort study of 647 subjects found that there was an inverse association between aspirin and the presence of AD, and that this was irrelevant to the aspirin dosage [27]. Various other observational, cohort, and case–control studies have supported the role of aspirin in the prevention of AD [32,43,44,46,58–62]. Overall, these cohort studies imply that aspirin may protect against AD, and that—especially in the elderly population—this may take place via a non-inflammatory pathway at lower doses of aspirin, as vascular changes are more common in elderly populations as compared to younger ones [46].

These findings were summarised in a recent meta-analysis and systematic review published in 2018 involving 10 studies of more than 24,418 patients. In contrast to individual

cohort studies, this meta-analysis—as well as an earlier meta-analysis in 2013—did not find any significant effect of aspirin use on incident AD [63,64]. However, this contrasts with previous meta-analyses in 2015 and 2009, which found that the risks of AD were lower in aspirin users (RR, 0.77; 95%CI, 0.63–0.95) compared with non-users [65].

Since 2018, the Alzheimer's Disease Neuroimaging Initiative longitudinal study published in 2021 found that aspirin was associated with slower cognitive decline in AD, but not but in patients with normal cognition or mild cognitive impairment [28].

The discrepancies between cohort studies and meta-analyses have a variety of potential causes, including eligibility criteria and included studies. The possibility of confounding and bias may be more prominent in observational studies, and selection bias is more likely in case–control studies. There is also a risk of recall bias, with information on aspirin use obtained from patient interviews and self-reporting in some of the cohort and case–control studies. This may ultimately bias the results in favour of the exposure in question [65]. Moreover, there may be lifestyle differences between the groups, including physical exercise, education level, and dietary patterns, all of which have been shown to affect cognitive decline [66]. In addition, the observed associations may differ based on the clinical stages at which aspirin was started. These issues need to be investigated further in future studies.

### 3.3. Aspirin and Cognitive Decline in Vascular Dementia (VD)

Vascular dementia is defined as "the result of infarction of the brain due to vascular disease, including hypertensive cerebrovascular disease. The infarcts are usually small but cumulative in their effect [67]". With vascular dementia being secondary to ischaemic milieu, low-dose aspirin's antiplatelet effects should theoretically result in reduced progression and reduced rates of vascular dementia. As such, there is some evidence of aspirin's preventative effect against cognitive decline in this population.

There has only been one randomised controlled trial investigating aspirin's effects on VD. Meyer carried out a randomised controlled trial of 70 patients with VD who either received 325 mg of aspirin daily or did not receive aspirin. After 1 year, patients on aspirin were found to have significantly better cognitive scores compared to the control group, and they also showed improvements in cerebral perfusion values [68]. Two other prospective cohort studies (the German ESTHER study, $n$ = 5258 [30]; and UK Biobank, $n$ = 305,394 [31]) have also found that low-dose aspirin could reduce the hazard for VD by 69% [37]. The benefit of low-dose aspirin in VD was additionally made more apparent in patients who used aspirin for more than 10 years (HR [95% CI]: 0.48 [0.42 to 0.56]), and this benefit was not seen in short-term users of aspirin [37]. These studies suggest that both low-dose and high-dose aspirin may reduce the decline in cognition in patients with VD. However, the Sydney Older Persons Study and the Cardiovascular Health Cognition Study both did not find that aspirin protected against vascular dementia [27,32].

Interestingly, some studies found that aspirin was in fact associated with a higher risk of vascular dementia. The Canadian Study of Health and Aging cohort study of 8623 dementia-free patients and the Rotterdam cohort found an increased risk of VD in aspirin users [33,69]. This may have been related to incident bias, with patients with VD more likely to be on aspirin. It may also be related to Neyman bias, with aspirin potentially reducing mortality rates in patients with vascular dementia and resulting in prolonged survival. Indeed, aspirin is known to reduce all-cause mortality in patients with cardiovascular risk factors, which is the profile of patients with vascular dementia [70], although the effects of aspirin on mortality specifically in vascular dementia have not yet been studied. That being said, aspirin has not been found to have a mortality benefit in all-cause dementia [19,22], with some studies instead suggesting that aspirin confers an increased mortality risk in patients with no cardiovascular disease [22].

To note, there are intrinsic difficulties in evaluating the effects of aspirin on vascular dementia by longitudinal means. Specifically, it is often diagnosed based on a background of cardiovascular risk factors, stroke, or myocardial infarction—a population that is likely to be on aspirin already and have other cardiovascular risk factors that may also play a role

in the development of vascular dementia [33,71]. Further studies should be carried out to clarify the role of aspirin in vascular dementia.

### 3.4. Aspirin and Cognitive Decline in Cerebral Small Vessel Disease (CSVD)

Cerebral small vessel disease (i.e., small arteries, arterioles, veins, and capillaries) is characterised by typical MRI findings including white matter changes, cerebral microbleeds, microinfarcts, and enlarged perivascular spaces [72]. It is known to be a major contributor to mixed dementia and vascular cognitive impairment [73,74], and patients often have a gradual progression of cognitive impairment that is associated with white matter changes [73,75]. Antiplatelet use with at least SAPT aspirin is well established as being a key aspect of the management of lacunar infarcts [76], and the SPS3 study found that a significant number of patients with lacunar infarcts had cognitive impairment [77]. However, the limited number of existing trials available have not yet proven the effectiveness of aspirin in preventing cognitive decline in this population.

The CHALLENGE trial (Comparison Study of Cilostazol and Aspirin on Changes in Volume of Cerebral Small Vessel Disease White Matter Changes) was a 104-week multicentre, double-blind, randomised controlled trial to compare the effects of cilostazol vs. aspirin in patients with CSVD diagnosed on imaging (i.e., moderate–severe white matter changes and at least one lacunar infarctions) [35]. They found that there were no significant differences in cognition or the progression of white matter changes between the aspirin and cilostazol groups over 2 years. There were also no differences in the effects of cilostazol or aspirin compared to the expected progression of white matter changes in CSVD [35,78]. The longitudinal Silence Study investigating the effects of aspirin vs. placebo on silent brain infarcts over 4 years similarly found that while aspirin protected against cardiovascular events, there was no difference in cognitive decline between the two groups [36]. At present, guidelines have not established the role of aspirin in the prevention of recurrent silent brain infarctions or symptomatic stroke in patients with silent brain infarction [79]; as such, evidence does not suggest a role of aspirin in preventing cognitive decline in patients with CVSD either. With the slow progression of CSVD, longer follow-up studies may be required to ascertain the longer-term implications of aspirin with respect to cognitive decline in patients with CSVD.

### 3.5. Aspirin and Dementia in Patients with Coronary Heart Disease (CHD)

Patients with CHD have an increased risk of recurrent vascular events, including cerebrovascular damage and cerebral ischaemia. This can result in cognitive impairment. As such, underlying vascular dysfunction is part of the neurodegenerative pathology in AD and VD [80,81], and there is evidence that aspirin may reduce dementia in patients with pre-existing CHD.

There is a complex interaction between the nervous and cardiovascular systems—the "heart-brain axis" [82]. Cerebrovascular diseases can result in changes on electrocardiogram and cardiac arrhythmias. The autonomic nervous system's sympathetic and parasympathetic systems—regulated by cerebral structures—also affect cardiac function [82]. Cognitive impairment itself may also affect patient compliance with medications and lifestyle modifications that would be important in preserving health and function. Moreover, with left ventricular failure, reduced cardiac output reduces cerebral flow, affects cerebrovascular reactivity, and increases the risk of cognitive decline [83], and both cardiovascular and some neurodegenerative diseases share similar risk factors. One cross-sectional description study found that up to 70% of NYHA class III or IV heart failure patients had at least mild cognitive impairment [84], with a systematic review of 2937 HF patients compared to 14,848 controls similarly finding that patients with heart failure had 1.6x more cognitive impairment compared to controls [85]. As such, aspirin can potentially act to prevent myocardial infarction and subsequent LV dysfunction that would otherwise lead to cognitive decline.

Whilst the association between stroke, AF, and cardiovascular disease has been better studied, the number of studies investigating the association between CHD and cognitive decline superficially is more limited. The longitudinal studies published so far have revealed mixed results. Some found that incident cardiovascular disease is associated with accelerated cognitive decline [86–90], whereas others did not [91,92].

A meta-analysis of two prospective cohort studies published in May 2022 (the German ESTHER study, *n* = 5258 [67]; and UK Biobank, *n* = 305,394 [68]) found that the use of low-dose ASA was weakly associated with decreased all-cause dementia incidence, but not with AD and VD incidence. However, after stratifying by CHD, it became apparent that only subjects with pre-existing CHD benefited strongly from low-dose ASA use [37]. This is also consistent with the findings of a longitudinal follow-up study from the JPAD trial (JPAD2) and the ASPREE trial, which did not find that aspirin reduced the risk of dementia or cognitive decline, as these studies excluded patients with CHD [20,22,39]. This suggests that the role of aspirin in reducing dementia in patients with pre-existing CHD should strongly be considered.

### 3.6. Aspirin and Dementia in Patients with Late-Onset Depression (LOD)

LOD is closely associated with dementia and is commonly found in patients with vascular disease [38,93]. The pathophysiology of depression includes a pro-inflammatory state with increased levels of IL-1, IL-6, and TNF-alpha [94,95]. Indeed, aspirin is associated with reduced levels of inflammatory markers associated with depression [96], via its inhibition of prostaglandin and other pro-inflammatory cytokines. Hence, aspirin's anti-inflammatory effects may help to reduce the neurological inflammation that is pertinent in depression.

Ya-Hsu Yang et al. carried out a population-based study of 6028 patients with LOD and 40,411 patients without LOD. They found that in patients with LOD, there was a lower incidence of incident dementia in patients on aspirin compared to those not on aspirin (hazard ratio = 0.734, 95% CI 0.641–0.841, *p* < 0.001). This finding was sustained after matching aspirin users with non-users by propensity scores (*p* = 0.022) [38]. However, other studies have suggested that aspirin may in fact confer an increased risk of depression [24,96], which may limit its potential use for the treatment of dementia in patients with depression.

### 3.7. Aspirin in Women

Sex differences in cardiovascular disease are well known, with cardiovascular disease being the leading cause of mortality for both sexes, but with men having a higher cardiovascular disease mortality rate [68]. As such, sex-specific studies have been carried out, and some have shown that aspirin confers a protective cardiovascular effect in women, reducing the risk of myocardial infarction and stroke [97–100]. A longitudinal follow-up of the original JPAD cohort study (JPAD2) was conducted on 2359 patients who had diabetes but no established cardiac disease [39]. It found that whilst overall aspirin did not significantly affect dementia, a significant reduction in the risk of dementia was seen in women, but not in men. Similarly, another Swedish trial of 681 women at high cardiovascular risk found significantly less dementia in women who were randomised to the aspirin group vs. the placebo group, but this difference was not seen in men [40]. The opposite was found in the Alzheimer's Disease Neuroimaging Initiative, which found that aspirin was associated with slower decline in male AD patients but not in female AD patients [63]. However, the study was also self-reported and did not account for lifestyle differences that could have confounded the results. A meta-analysis of two cohort studies (the German ESTHER study [30] and UK Biobank [31]) found that low-dose aspirin was associated with a decreased risk of all-cause dementia and VD in males, but not in females [37].

Studies on aspirin's effects on cognition specifically in women are more limited. The Women's Health Study was a randomised, double blind, placebo-controlled cohort study of low-dose aspirin in healthy women >65 years old. A total of 6377 participants were randomly assigned to receive low-dose aspirin or placebo (mean age at baseline cognitive

assessment = 71.8 years). A sub-study analysis found that women assigned aspirin did not do better than those assigned placebos over a mean of 5.6 years on either global cognitive screening or memory tests, although they did find some evidence for reduced decline in category (semantic) fluency by 20% in the aspirin group (relative risk 0.80, 0.67–0.97). Aspirin was associated with less cognitive decline than placebo in 2 of the 14 subgroups (i.e., current smokers and participants with hyperlipidaemia). Adverse events were not reported [41]. Overall, the study found that long-term use of low-dose aspirin did not convey a benefit for cognition in generally healthy women > 65 years old. A subgroup analysis of the ASPREE trial also did not find any significant differences between the aspirin and placebo groups, including the subgroup defined by sex [22].

Similarly, Kang et al.'s observational study of 13,255 women >70 years old found that long-term use of low-dose aspirin did not have a significant association with cognitive decline in women [42]. These two studies suggest that low-dose aspirin, at 4–15 years of therapy, does not affect cognitive decline. It should be noted that both of these studies used low-dose aspirin with its predominantly antiplatelet therapy at this dose. Whether or not aspirin's anti-inflammatory properties at higher doses may affect cognitive decline in women has not yet been investigated.

Differences in study outcomes may be affected by differences in baseline demographics. The Women's Health Study and the observational study by Kang both used healthy females, excluding women with cardiovascular disease, and did not find a gender-dependent effect of aspirin. In contrast, the Swedish trial investigated aspirin's effects on women at high cardiovascular risk and found a significant effect. However, the JPAD2 trial excluded patients with established cardiovascular disease but also found a gender difference in outcomes.

Overall, the role of aspirin in reducing cognitive decline in women with cardiovascular disease has shown mixed results, and this may also be related to the low doses of aspirin used in these studies and, therefore, its predominantly antiplatelet effect, as compared to its anti-inflammatory effect at higher doses. The small number of studies on gender differences require further investigation for more robust conclusions.

### 3.8. Impact of Age

Age itself is a significant risk factor for cognitive decline, and the effects of aspirin on cognitive decline may vary based on age. Most trials recruited elderly patients and found limited benefit of aspirin in preventing cognitive decline. Age has been shown to affect treatment outcomes for other similar medications: The Medical Research Council Treatment Trial of Hypertension in Older Adults found that whilst NSAIDs protected against cognitive decline, this effect was not seen in patients >74 years old [27]. The Baltimore Longitudinal Study of Aging supports a hypothesis that the effect of aspirin may also be dependent on age, with the recruited subjects being younger at <65 years old, and finding a reduced risk of AD with aspirin [46]. However, other studies have found that aspirin is beneficial in older populations [37,45]. Nonetheless, with cognitive decline being most prevalent in elderly populations, the role of aspirin in reducing cognitive decline in younger populations may be of limited value.

### 3.9. The Role of High-Dose Aspirin

The dose of aspirin may affect the outcomes. At higher doses (i.e., 1.2–3.6 g/day), aspirin predominantly acts on COX-2 rather than COX-1, inhibiting the conversion of arachidonic acid to prostaglandins. Prostaglandins play a crucial role in CNS neuroinflammation as well as in the development of Alzheimer's disease [53]. In Alzheimer's disease, the extracellular deposition of amyloid β-peptide, intracellular neurofibrillary tangle deposition, and neuronal loss induce inflammatory pathways that result in accumulation of prostaglandins and other inflammatory mediators [53]. This results in neuronal degeneration. As such, increased levels of inflammatory mediators such as prostaglandin in the CSF and the brain parenchyma are associated with cognitive impairment [51,52,54,55].

Studies have shown that direct application of prostaglandin E2 to the brain parenchyma increases memory deficits [101], and COX-2 overexpression also impairs neuronal function [102]. With COX-2 being induced by inflammatory processes in the microglia and endothelium [103,104], some studies have found increased levels of prostaglandins in patients with AD [105]. However, Combrinck's longitudinal study found that whilst PGE2 was initially elevated, its levels subsequently declined as memory declined [106]. This suggests a more complex role of COX-2 and prostaglandins, with evidence that COX-2-derived PGE2 is an essential aspect of normal brain function and memory [107–109].

Extensive studies investigating the role of COX-2 inhibition have been carried out using NSAIDS, all largely congruent with its lack of effect in reducing cognitive decline [110–115]. However, fewer studies have been conducted on the COX-2-inhibitory effects of high-dose aspirin. A Swedish longitudinal study of 702 patients >80 years old found that high-dose aspirin resulted in lower prevalence of AD and better cognitive function compared to non-users [43]. The Sydney Older Persons Study of 647 subjects found that whilst aspirin reduced the risk of developing Alzheimer's disease, there was no difference in outcomes between low and high doses [27]. As such, it is inconclusive whether high-dose aspirin could protect against cognitive impairment or dementia.

### 3.10. Aspirin and Cerebral Haemorrhage

In addition to gastroduodenal toxicity, aspirin confers a risk of bleeding, including risk of cerebral haemorrhage [22]. Cerebral microbleeds are associated with cognitive decline and both vascular and Alzheimer's dementia, and they are a hallmark of diffuse neurodegenerative damage [34,116]. Aspirin may increase the risk of cerebral haemorrhage, both through its effects on the inhibition of platelet aggregation and via impairment of platelet thromboxane A2 synthesis. In addition, studies have found that aspirin aggravates microbleeds in patients with cerebral amyloid angiopathy [34,117].

Both the AD2000 trial and the ASPREE trial found that the risk of serious bleeding was increased in elderly patients on aspirin compared to placebo (RR = 4.4, 95% CI = 1.5–12.8, and hazard ratio, 1.38; 95% CI, 1.18 to 1.62, respectively) [22,23]. That being said, a Taiwanese registry found that dementia itself is significantly associated with ICH. In this study, patients with AD had a hazard ratio of 2.02 (1.10–3.72) for ICH compared to matched controls without AD. The study also found that aspirin may potentiate the risk of ICH in AD, with patients with AD on aspirin having a hazard ratio of 2.22 (1.07–4.62) for ICH compared to matched controls [118]. This suggests that the extra risk of intracranial haemorrhage in individuals with dementia on aspirin is around one case per thousand people. This finding was similar to that of the Rotterdam Study, which found that older adults on aspirin were 2.7 times more susceptible to microbleeds on MRI [116]. Multiple other studies have also found antiplatelet use to be associated with increased risk of cerebral microbleeds [119–122]. However, a Japanese cohort of patients with ICH found that cerebral microbleeds were not associated with antiplatelet use [123]. It seems that the potential benefit of aspirin in reducing dementia via reduced vascular events may be counterbalanced by a twofold increase in the risk of major bleeding [19]. However, these data do not include subclinical events such as microbleeds, microinfarcts, silent brain infarcts, or unreported TIAs. Currently, the ASPREE-NEURO study is underway to determine the effects of aspirin on the risk on cerebral microbleed development over 3 years [124].

### 3.11. Dual- or Triple-Antiplatelet Therapy: Aspirin in Combination with Dipyridamole or Clopidogrel

Dipyridamole used with aspirin has been shown to be neuroprotective in cell cultures of neurons and in rats with embolic stroke [125–127]. However, this has not translated into clinical benefit for secondary prevention in humans. Treatment with clopidogrel has also not been shown to have neuroprotective effects in animal models [128]. As such, we examined the evidence for dual antiplatelet therapy (DAPT) or triple antiplatelet therapy in preventing cognitive decline/dementia in humans.

The PRoFESS trial (Prevention Regimen for Effectively Avoiding Second Strokes) randomised 20,332 patients (695 sites in 35 countries) with previous ischaemic stroke into either 25 mg of aspirin (ASA) and 200 mg of extended-release dipyridamole twice a day or 75 mg of clopidogrel once a day, and 80 mg of either telmisartan or placebo once per day. They postulated that these drugs could reduce the size of thromboembolism and possessed intrinsic neuroprotective properties that could potentially limit brain injury post-stroke. Between all treatment groups, there were no significant differences in the median MMSE scores or rates of dementia after a median of 2.4 years, suggesting no additional benefit of DAPT compared to placebo [129]. That being said, the PRoFESS trial was not designed or powered to test the neuroprotection hypothesis [130].

The SPS3 (Secondary Prevention of Small Subcortical Strokes) randomised controlled trial of 2916 patients with recent symptomatic lacunar infarcts also did not find that cognition was affected by DAPT as compared to single antiplatelet therapy [131]. It should be noted that there was variability in the definition of cognitive impairment and in the time from the onset of stroke to the initiation of therapy. One reason for this may have been trial design, with reduced risk of recurrent stroke in the patient population compared to historical data, as well as tighter risk factor control [132]. Studies on stringent blood pressure control suggest that well-controlled blood pressure can reduce the risk of cognitive decline [133–135]. As such, whether or not antiplatelet therapy is effective in reducing cognitive decline may also be affected by blood pressure control, and its effects on cognitive impairment may be less obvious in the context of well-controlled blood pressure. The study also did not evaluate the effects of DAPT compared to placebo.

There is still debate on this matter, with a randomised controlled trial by Chun Wang et al. of 574 patients with LAA stroke finding that there was less early neurological deterioration in patients on DAPT compared to aspirin alone after 30 days [136]. This suggests that DAPT may be beneficial in preventing early neurological deterioration in certain subgroups, such as those with acute large-artery strokes. Similarly, a longitudinal study of 4413 patients on the community-based South London Stroke Register found that aspirin and dipyridamole reduced the risk of cognitive impairment (relative risk, 0.8 [95% confidence interval, 0.68–1.01]) [133].

Some have gone a step further in investigating the role of triple therapy. The TARDIS (Triple Antiplatelets for Reducing Dependency after Ischaemic Stroke) randomised controlled trial investigated the role of triple antiplatelets (i.e., combined aspirin, clopidogrel, and dipyridamole) vs. guideline therapy (either combined aspirin and dipyridamole, or clopidogrel alone) in patients with acute non-cardioembolic ischaemic stroke or TIA [137]. It tested whether short-term intensive antiplatelet therapy was safe and effective in reducing the early risk of recurrent stroke, and the study had a secondary endpoint of cognitive impairment. It did not find that there was any difference in cognitive impairment between standard guideline therapy and triple therapy. This trial adds to the evidence from other trials that antiplatelets have limited utility in reducing cognitive decline—whether it is a single, dual, or triple antiplatelet therapy. That being said, the intervention in the TARDIS trial was only 30 days; to fully evaluate its effects on cognitive decline may require a longer trial period.

In addition, some trials had higher rates of cerebral haemorrhage with a multiple-antiplatelet regimen. The PRoFESS trial found there to be more haemorrhagic strokes with aspirin combined with extended-release dipyridamole compared to clopidogrel alone, although there was no significant difference in the risk of fatal or disabling stroke [138]. The SPS3 and TARDIS trials also had higher rates of cerebral microbleeds and cerebral haemorrhage [137,139]. Amongst the studies that suggested a beneficial effect of DAPT on cognitive impairment, Chun Wang et al.'s RCT did not find an significant increase in cerebral haemorrhage in patients on DAPT [136], although DAPT is well known to increase the risk of bleeding events [140]. Hence, the risk vs. benefits of a multiple-antiplatelet regimen for cognitive impairment should be carefully considered, in view of conflicting evidence of its efficacy as well as potentially increased risk of bleeds.

*3.12. Limitations*

With large trials so far not showing any significant effect of aspirin in preventing cognitive decline, and with a possible increased risk of aspirin in contributing to small microbleeds that may increase cognitive impairment, at this juncture there is likely a limited role of aspirin in preventing cognitive decline. Further studies should be conducted on whether higher doses of aspirin can be considered, as most trials used low-dose aspirin (75–100 mg), which has a predominantly antiplatelet effect rather than an anti-inflammatory effect. Aspirin's antiplatelet effect may therefore require a longer duration to observe any benefit [45], especially taking into consideration that dementia can take 15–25 years to develop [141]. Most trials were powered for 3–7 years, which may be too short a duration to sufficiently evaluate the potential benefits of low-dose aspirin. Longer-duration trials could be considered; however, compliance with a longer duration of therapy with questionable benefits and a potentially adverse risk profile may be problematic in the group of patients that this therapy is likely to benefit, as they are often elderly and have a shorter life expectancy.

There was also significant variation in the tools used to assess cognitive decline, as well as variability in the definition of cognitive impairment. Some studies had the majority of participants scoring full marks in their baseline cognitive test, suggesting that they had hit a ceiling. Whether or not they could have experienced a degree of cognitive decline insufficient to affect their final test scores remains to be evaluated. There was also significant variability in the baseline characteristics of patients between studies, including their degree of cardiovascular risk factors, presence of cognitive impairment/dementia prior to the intervention, and presence of stroke/TIA, all of which could affect the pace of cognitive decline or dementia.

**4. Conclusions**

Overall, the benefits of aspirin in preventing dementia and cognitive decline remain inconclusive, with a large variety of conflicting evidence. The majority of trials did not find undeniable evidence of aspirin's role in preventing cognitive decline, although there was some evidence for its role in vascular dementia and coronary heart disease patients. Whilst the bulk of the cohort studies looked promising, all randomised controlled trials except one on VD have failed to demonstrate that aspirin plays a significant role in preventing cognitive decline or dementia. The clinician must also consider the side effects and risks of bleeding with aspirin, especially in the population of more elderly patients for whom aspirin is likely to provide a benefit. Nonetheless, aspirin is a commonly used drug for both primary and secondary prevention of cardiovascular disease, and whether it may be beneficial in certain subgroups of patients (such as CHD, VD, and CSVD) warrants further investigation and more robust studies, especially with the general paucity of RCTs in these subgroups.

**Author Contributions:** Conceptualization, E.H.T. and C.-H.S.; methodology, E.H.T. and C.-H.S.; validation, E.H.T. and C.-H.S.; formal analysis, E.H.T.; investigation, E.H.T.; resources, E.H.T.; data curation, E.H.T.; writing—original draft preparation, E.H.T.; writing—review and editing, E.H.T., E.C.Y.L., C.-Y.Y. and T.Y.W.L.; supervision, C.-H.S. All authors have read and agreed to the published version of the manuscript.

**Funding:** This research received no external funding.

**Institutional Review Board Statement:** Not applicable.

**Informed Consent Statement:** Not applicable.

**Data Availability Statement:** Data sharing not applicable. No new data were created or analyzed in this study. Data sharing is not applicable to this article.

**Conflicts of Interest:** The authors declare no conflict of interest.

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
