# Peer review of "Aspirin Therapy, Cognitive Impairment, and Dementia—A Review"

_futurepharmacol, doi:10.3390/futurepharmacol3010011_

Round 1

Reviewer 1 Report

Mild cognitive impairment MCI and dementia (different types) -   are very frequent in patients with cardiovascular risk factors.

Aspirin is used drug for both primary and secondary prevention in patients with cardiovascular diseases.

The anti-platelet role of low dose of acetylsalicylic acid (75-300 mg/day with COX1 inhibition COX-1 and secondary inhibition of thromboxane A2 synthesis) and the anti-inflammatory role of high dose of acetylsalicylic acid in in preventing cognitive decline are evaluated by the authors.

Studies results are divergent, the majority of trials did not bring undeniable evidence of preventing role of aspirin in cognitive decline, with exception of vascular dementia, cerebral small vessel disease and coronary heart disease patients.

Reviewer 2 Report

The review is well written and easy to follow.

The format is well organized and described in each impact of aspirin with cognitive decline and dementia.

The references are updated.
